# Proprioceptive and cutaneous sensations in humans elicited by intracortical microstimulation

Michelle Armenta Salas[1,2†], Luke Bashford[1,2†], Spencer Kellis[1,2,3,4†], Matiar Jafari[1,2,5], HyeongChan Jo[1,2], Daniel Kramer[3,4], Kathleen Shanfield[6], Kelsie Pejsa[1,2], Brian Lee[3,4], Charles Y Liu[3,4,6], Richard A Andersen[1,2]*

[1]Department of Biology and Biological Engineering, California Institute of Technology, Pasadena, United States; [2]T & C Chen Brain-Machine Interface Center, California Institute of Technology, Pasadena, United States; [3]USC Neurorestoration Center, Keck School of Medicine of USC, Los Angeles, United States; [4]Department of Neurological Surgery, Keck School of Medicine of USC, Los Angeles, United States; [5]UCLA-Caltech Medical Scientist Training Program, Los Angeles, United States; [6]Rancho Los Amigos National Rehabilitation Center, Downey, United States

**Abstract** Pioneering work with nonhuman primates and recent human studies established intracortical microstimulation (ICMS) in primary somatosensory cortex (S1) as a method of inducing discriminable artificial sensation. However, these artificial sensations do not yet provide the breadth of cutaneous and proprioceptive percepts available through natural stimulation. In a tetraplegic human with two microelectrode arrays implanted in S1, we report replicable elicitations of sensations in both the cutaneous and proprioceptive modalities localized to the contralateral arm, dependent on both amplitude and frequency of stimulation. Furthermore, we found a subset of electrodes that exhibited multimodal properties, and that proprioceptive percepts on these electrodes were associated with higher amplitudes, irrespective of the frequency. These novel results demonstrate the ability to provide naturalistic percepts through ICMS that can more closely mimic the body's natural physiological capabilities. Furthermore, delivering both cutaneous and proprioceptive sensations through artificial somatosensory feedback could improve performance and embodiment in brain-machine interfaces.
DOI: https://doi.org/10.7554/eLife.32904.001

*For correspondence:
andersen@vis.caltech.edu

†These authors contributed equally to this work

Competing interests: The authors declare that no competing interests exist.

## Introduction

The absence of somatosensation profoundly diminishes a person's ability to move and interact within their environment (*Cole and Cole, 1995*; *Sainburg et al., 1993*). Even with intact vision and hearing, which can provide sensory information about body position, movement, and interaction, basic behaviors such as walking or reach-and-grasp require substantially greater cognitive load without somatosensory feedback. The severity of these deficits underscores how deeply integrated cutaneous and proprioceptive somatosensations are in the neural control of movement, and motivates the problem of restoring sensation when it is missing. However, the complexity of the somatosensory circuit, and the difficulty of writing information into this circuit with sufficient integrity, have posed significant challenges. Recent advances in brain-machine interface (BMI) technology have led to renewed efforts in this area, under the hypothesis that providing closed-loop motor-sensory control and feedback pathways could lead to vital increases in performance (*Bensmaia and Miller, 2014*). Intracortical microstimulation (ICMS) is a promising technique for implementing a return path in which electrical stimuli are written directly into the somatosensory cortex through implanted

**eLife digest** Nerves throughout the body send information about touch, temperature, body position and pain through the spinal cord to the brain. A part of the brain called the somatosensory cortex processes this information. Spinal cord injuries disrupt these messages. Even though the somatosensory cortex has not been damaged, sensation is lost for the affected body areas. No treatment exists to repair the spinal cord so the loss of sensation is permanent.

Applying electricity to the somatosensory cortex can produce artificial sensations. Scientists are testing this approach to restore a sense of touch for people with spinal cord injury. Early experiments show that using different patterns of electrical stimulation generates unnatural sensations in different body parts. People receiving the stimulation describe it as tingling or shocks. Scientists wonder if they can improve the technique to mimic feelings like touch or body position to make it easier for people with a spinal injury to move or use prostheses.

Now, Armenta Salas et al. generated more natural sensations in a person with a spinal cord injury. Instead of taking the usual approach of delivering large currents to the surface of cortex, they inserted small electrodes into the inside of the cortex to stimulate it with small currents. In the experiments, electrodes were implanted in the somatosensory cortex of a volunteer who had lost the use of his limbs and torso because of a spinal injury. Armenta Salas et al. applied different patterns of electrical stimuli and the volunteer reported what they felt like. The patient described sensations like a pinch or squeeze in the forearm or upper arm with certain patterns. In some cases, the patient reported the sensation of the arm moving with stronger electrical currents.

The experiments show that electrical stimulation of the brain can recreate some natural sensations. These sensations could help patients using robotic or prosthetic arms become more dexterous. It might also help patients view artificial limbs as part of their bodies, which could improve their sense of wellbeing.

DOI: https://doi.org/10.7554/eLife.32904.002

electrode arrays. Non-human primates (NHPs) successfully incorporated ICMS information to perform discrimination, detection tasks (*Romo et al., 1998*; *Romo et al., 2000*; *Tabot et al., 2013*; *Dadarlat et al., 2015*) and as sensory feedback for brain control in BMI tasks (*O'Doherty et al., 2011*; *Klaes et al., 2014*), and recent human studies have provided insight into the feeling and perception of the sensations produced through ICMS (*Flesher et al., 2016*). However, qualities ascribed by human subjects to these sensations (e.g., 'tingling' or 'buzzing') have been mostly artificial in nature (*Johnson et al., 2013*; *Flesher et al., 2016*), and it is as yet unclear what range of sensations could be elicited through ICMS. Here, we present novel findings from two experiments: one which tested each electrode over a range of amplitudes with fixed frequency, and one which tested a subset of electrodes over a range of amplitudes and frequencies. We found reliable elicitation of natural cutaneous and proprioceptive sensations spanning a range of stimulus amplitudes and frequencies, obtained from stimulation in S1 of a single human subject (participant FG, *Figure 1*; see Materials and methods) with a C5-level spinal cord injury. We further show that current amplitude, not frequency, of the electrical stimulus differentiates the modality (i.e., cutaneous or proprioceptive) of the elicited percept at some stimulation sites.

## Results and discussion

In Experiment 1, over an eight-week period, electrical stimuli were tested across a range of current amplitudes between 20–100 µA, with pulse frequency held constant at 150 Hz (see Materials and methods). Stimulation through 46/96 electrodes (48%) prompted at least one response, and there were in total 381 reported sensations out of 1229 non-catch trials (see Materials and methods). There was weak correlation between the number of electrodes that elicited a sensation and the current amplitude (r = 0.34, p=0.42, Pearson linear correlation). Additionally, we found no correlation between electrode impedance and the likelihood of elicited percepts (p=0.80, Pearson linear correlation coefficient), pooling all electrode responses over all days. Furthermore, there was no significant difference in the aggregate impedances of either electrodes that produced or did not produce percepts (p=0.707, Kolmogorov-Smirnov two-sample test). No false positives were reported in any

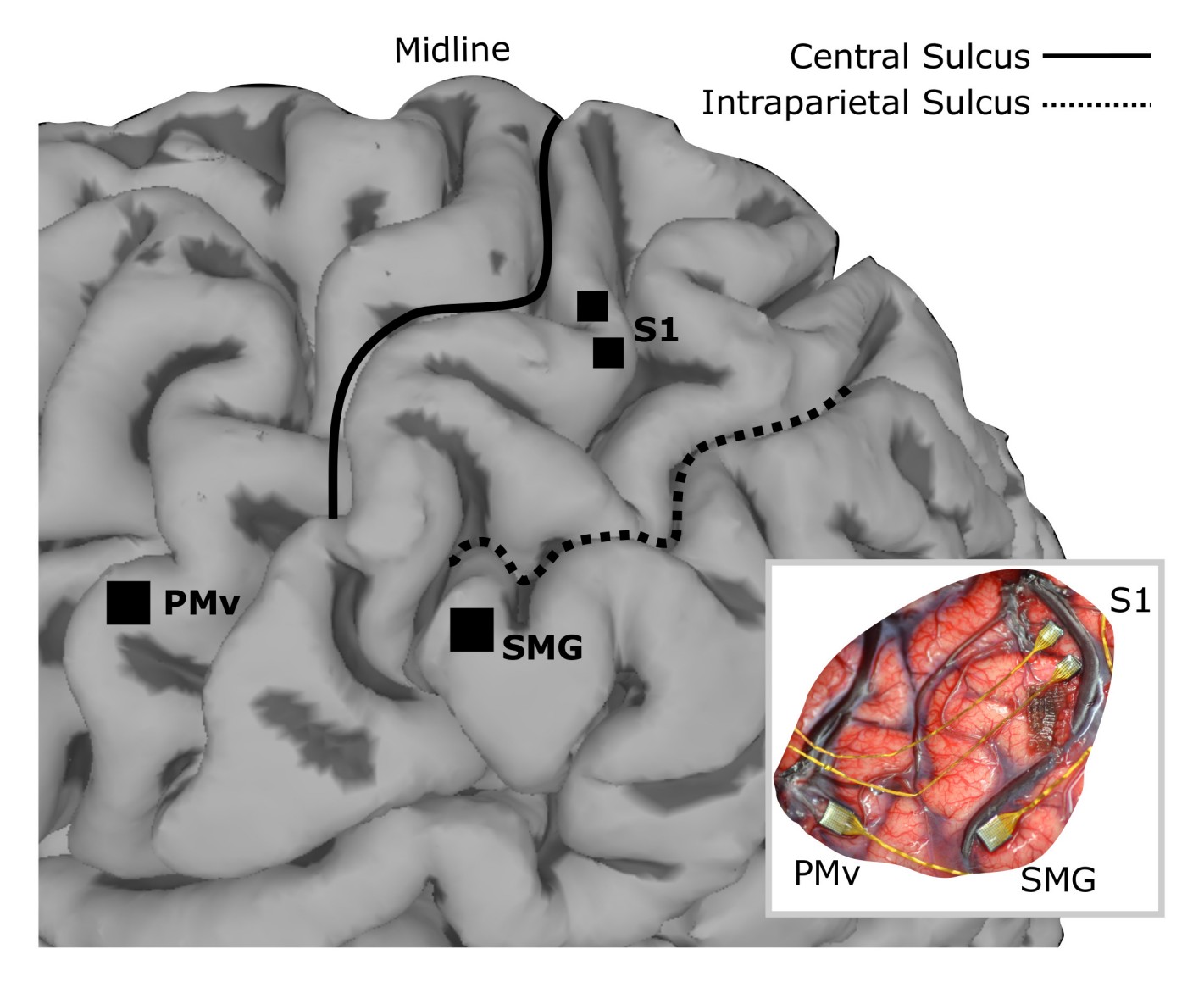

**Figure 1.** Array implant locations on rendered MRI image of the left hemisphere of FG. 96-channel microelectrode arrays were implanted into ventral premotor cortex (PMv) and supramarginal gyrus (SMG), and two 48-channel stimulating arrays were implanted into primary somatosensory cortex (S1). The insert shows the in situ array locations.

DOI: https://doi.org/10.7554/eLife.32904.003

catch trials, and we found no effect of trial history in the proportion of reported sensations during stimulation (see Materials and methods). The stimulation did not trigger any painful sensations, and no adverse events occurred during any of the sessions.

Receptive fields along the upper arm, forearm and hand corresponded to coarse somatotopical organization in the corresponding stimulation sites. *Figure 2* shows the most frequently reported receptive field and sensation modality for each electrode across all trials. Of the 46 electrodes with responses, 32 evoked percepts in the upper arm, 18 in the forearm, and two in the hand (palmar surface of digits and a finger pad). In agreement with previous reports, stimulation could produce percepts with variably-sized receptive fields in different electrodes (*Flesher et al., 2016*). For the majority of electrodes (24/46), receptive fields were reported in the same body region (i.e. upper arm or forearm) or in the same plane (i.e. anterior or posterior) across all tested amplitudes. Coarse somatotopy was present between the medial and lateral arrays (*Figure 2B*); the medial array was

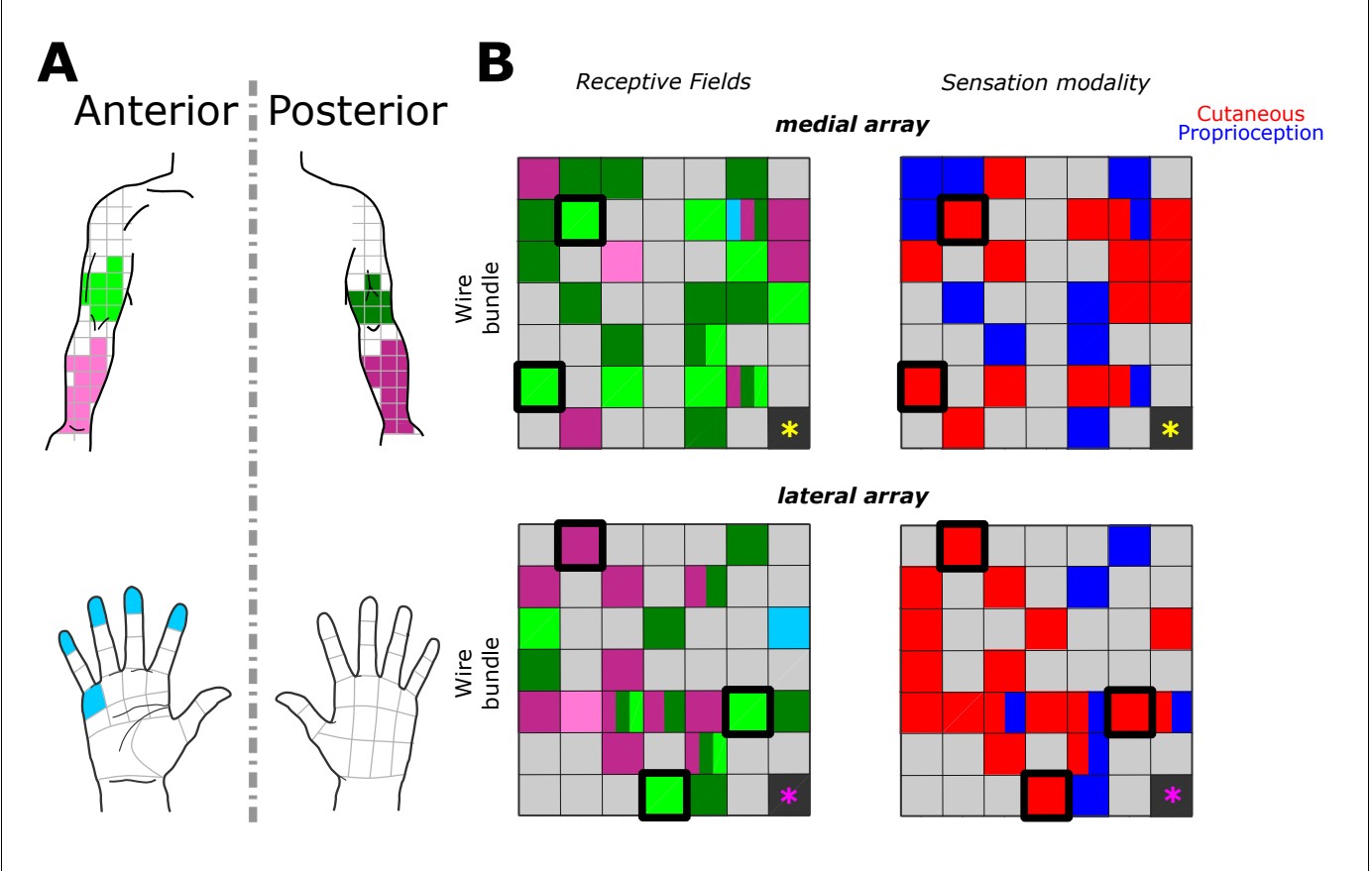

**Figure 2.** Receptive fields and sensation modality across all amplitude mapping experiments. (A) Receptive field location on anterior (lighter shades) and posterior (darker shades) planes of the right upper arm (green), forearm (pink), and hand (cyan). Grid is the same that the subject referenced during the experiment. (B) Schematic of the two electrode arrays implanted over S1 (*Figure 1*). Left side panels display the reported receptive fields at each electrode location, and right side panels display the sensation modality (cutaneous - red, proprioceptive - blue). Light gray boxes show electrodes with no reported sensation, while dark gray boxes represent reference channels which are not used in recording. The five electrodes with a thick black outline represent the subset tested in the additional parameter-wide mapping task. Yellow and magenta asterisks mark the inferior-posterior corner of the implants, for the medial and lateral arrays respectively.

DOI: https://doi.org/10.7554/eLife.32904.004

more likely to have reliable receptive fields in the anterior upper arm (46% of medial-array receptive fields), while stimulation on the lateral array induced sensation more frequently on the posterior forearm (51% of lateral-array receptive fields). However, there was no clear somatotopical organization within each array as previously reported (*Kim et al., 2015a*; *Kaas, 1983*; *Flesher et al., 2016*). The coarse somatotopy found across arrays but not within arrays, could be due to the small area of cortex sampled by the implants, and the fact that the implants predominantly covered upper arm and forearm, areas with a less established somatotopic map (*Kaas et al., 1979*; *Kaas, 1983*). Another plausible explanation is that the topography in somatosensory cortex has been remapped after injury (*Kaas et al., 1983*; *Florence et al., 1998*; *Moore et al., 2000*)

FG has reported a wealth of qualitative sensations induced by ICMS (*Table 1*). Unlike paresthetic sensations experienced post-injury, these naturalistic responses were broadly characterized as *cutaneous* (e.g. squeeze) or *proprioceptive* (e.g. rightward movement), and as being subjectively similar to sensations experienced prior to injury. At his own discretion, the subject used single-word descriptors to characterize the perceived sensations as accurately as possible. Single-word descriptors have the advantage that they can be compared across large data sets or subjects. However, as experimental advances continue to push the capabilities of ICMS, responses could become more complex and future studies might benefit from more structured descriptors, which take into consideration the complexity of these sensory experiences (*Darie et al., 2017*).

**Table 1.** Descriptions of the most prevalent sensations by percentage of total responses.
Entries cover 90% of 381 reported sensations, with the final 10% comprising a mixture of other naturalistic cutaneous and propriocep-
tive descriptors. Each sensation is accompanied by the mode and 25th-75th percentiles in the distribution of amplitudes that elicited
each sensation, and by the same quantities for the perceived reported intensities (on a scale of 1 [weak] to 10 [strong]).

| Description | % Total Sensations (381 total) | Amplitude μA (mode) | Amplitude μA (25th, 75th percentile) | Intensity (mode) | Intensity (25th, 75th percentile) |
|---|---|---|---|---|---|
| Squeeze | 24.9 | 40 | 40, 87.5 | 7 | 4, 7 |
| Tap | 17.3 | 70 | 40, 80 | 1 | 1, 4 |
| Right movement | 9.7 | 90 | 55, 90 | 1 | 1, 3 |
| Vibration | 8.1 | 40 | 40, 90 | 2 | 2, 3 |
| Blowing | 6.6 | 60 | 30, 80 | 1 | 1, 2 |
| Forward Movement | 5.8 | 70 | 40, 80 | 1 | 1, 4 |
| Pinch | 5.5 | 40 | 40, 90 | 3 | 3, 6 |
| Press | 5.0 | 40 | 40, 70 | 7 | 4, 7 |
| Upward Movement | 3.9 | 70 | 70, 85 | 1 | 1.25, 4 |
| Goosebumps | 3.1 | 100 | 60, 90 | 5 | 2, 5 |

DOI: https://doi.org/10.7554/eLife.32904.005

We found that 18 electrodes had cutaneous-only responses across all tested current amplitudes,
while six electrodes had proprioceptive-only responses; the rest of the electrodes (22/46) had mixed
responses, where the perceived modality (cutaneous or proprioceptive) varied as stimulus parame-
ters changed. Of these mixed-response electrodes, 45% evoked mostly cutaneous sensations, 32%
evoked mostly proprioceptive sensations, and 23% had an equal number of cutaneous and proprio-
ceptive sensations (*Figure 2B*). This pattern of cutaneous and proprioceptive evoked sensations
complements recent reports of multimodal (i.e. cutaneous and proprioceptive) neurons throughout
S1 (*Yau et al., 2016*; *Kim et al., 2015b*). While prior single-unit experiments have defined maps
from single neurons to specific unimodal receptive fields (*Kaas et al., 1979*; *Kaas, 1983*;
*Friedman et al., 2004*; *Romo et al., 2000*), the above results suggest that more than one variable
may be represented when mapping with ICMS. This finding may be the product of different mecha-
nisms by which receptive fields are observed through recording versus stimulation, and could be an
important topic for future work. We found a significant difference between the amplitudes that eli-
cited cutaneous or proprioceptive responses, with the distribution of proprioceptive responses
skewed towards higher amplitudes (*Figure 3A*), when pooling across all electrodes and amplitudes
that produced a sensation (p=0.039, Kruskal-Wallis nonparametric ANOVA, $\chi^2(1,378)=4.41$, proprio-
ceptive responses N = 79, cutaneous responses N = 302). To assess consistent current delivery
across all electrodes, we measured electrode impedance at the beginning of every session and
found no significant difference when comparing proprioceptive or cutaneous responses (p=0.237,
$\chi^2(1,378)=1.39$) and, furthermore, we found no significant difference between the impedance of pro-
prioceptive- and cutaneous-only (p=0.922, $\chi^2(1,155)=0.01$) or mixed-response electrodes (p=0.372,
$\chi^2(1,221)=0.8$). To account for potential bias from an uneven distribution of responses across ampli-
tudes, we compared the proportion of proprioceptive and cutaneous responses in a bootstrapped
resampling (N = 10000), in which each repetition drew 15 responses at each amplitude from all data
pooled across days (*Figure 3B*). We observed a clear relationship between the number of proprio-
ceptive and cutaneous responses and stimulation amplitudes, measured through overall positive
slopes in the 1st-order polynomial fit at each iteration for proprioceptive responses, and negative
slopes for cutaneous responses (*Figure 3C*).

Experiment 2 tested a subset of 5 electrodes with robust responses across all tested amplitudes
in Experiment 1 (*Figure 2B*, *Figure 3D*). In a pseudorandomly-interleaved fashion, we stimulated
each electrode with five amplitudes (range 20 to 100 μA) at six different frequencies (range 50 to
300 Hz) over the course of three consecutive days (see Materials and methods). We reproduced the
effect of amplitude on sensation modality, either proprioceptive or cutaneous, when pooling across
all responses (p=$2\times10^{-5}$, $\chi^2(1323)= 18.17$, *Figure 3E*). Similar to the main mapping task, we did not

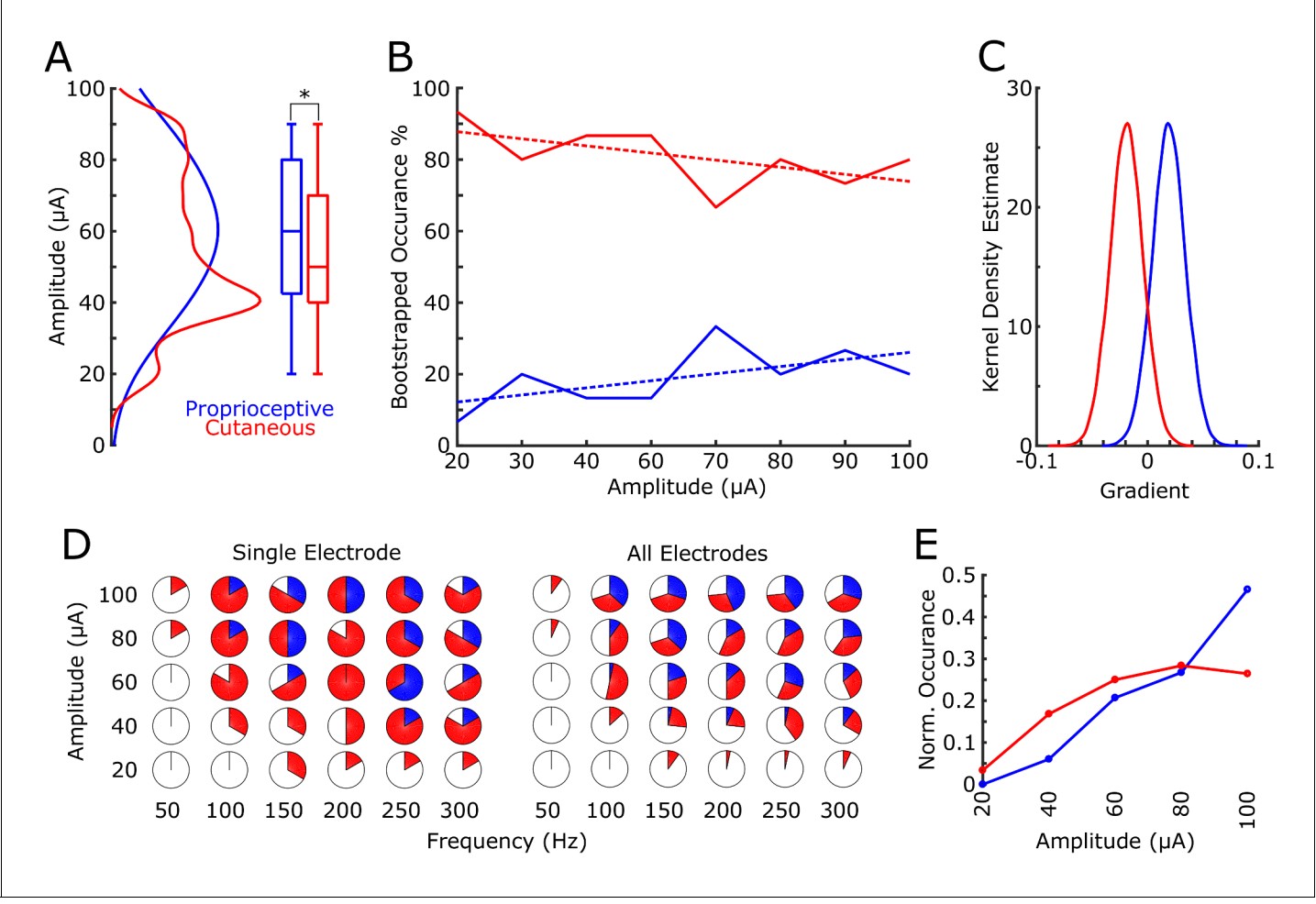

**Figure 3.** Proprioceptive and cutaneous responses. (A) Kernel density estimate and box plot showing the difference in the distribution of amplitudes associated with each report of proprioceptive (blue) or cutaneous (red) responses. (B) The median percentage of responses in the bootstrapped sample (solid line) for proprioceptive and cutaneous responses at each amplitude tested. Dashed line shows 1st-order polynomial fit. (C) Kernel density estimates of the distribution of slopes from 1st-order polynomial fits in each bootstrap iteration. (D) Pie charts show the percentage of total stimulations of responses for the subset of electrodes tested over a range of both current amplitudes and pulse frequencies. The left panel shows an individual example electrode (six trials per combination of amplitude and frequency) and the right panel shows data pooled over all five electrodes (30 total stimulations per combination). The percentage of no response (white), proprioceptive (blue) or cutaneous (red) are shown. (E) A normalized histogram of proprioceptive (blue) and cutaneous (red) responses at each of the amplitudes tested in experiment 2.
DOI: https://doi.org/10.7554/eLife.32904.006

find any significant effect on modality due to electrode impedance (p=0.305, $\chi^2$(1323)=0.8). Furthermore, there was no significance when testing the effect of frequency in eliciting proprioceptive or cutaneous responses (p=0.22, $\chi^2$(1323)= 1.48).

This amplitude-specific effect on sensation modality is perhaps surprising given the more commonly observed effect of frequency and pulse-width modulation on sensation in the periphery (*Graczyk et al., 2016*). Although there is evidence of tactile and proprioceptive inputs co-modulating S1 firing activity (*Kim et al., 2015b*), we are unaware of any reported effect of amplitude or frequency thresholding for different sensory modalities in the CNS. Proprioceptive sensations are commonly thought to derive from activity in areas 2 or 3a, while cutaneous sensations more likely correspond to activity in areas 3b and 1. From topographical features, we estimate our implants lie in area 1; however, with evidence of interindividual variability in the microstructural organization within S1 (*Geyer et al., 1999*), and the potential for functional reorganization after injury (*Kaas et al., 1983*; *Florence et al., 1998*), it is possible that higher current amplitudes could increase the effective range of stimulation to include sensory areas 3a or 2. Moreover, given the

receptive fields activated during stimulation, the two implants are well within the arm and forearm regions of S1, which might receive a larger ratio of proprioceptive-to-cutaneous signals than hand regions (*McKenna et al., 1982*), making it more likely to activate these different modalities with ICMS.

FG also provided subjective measures of sensation intensity and duration. Sensation intensity was ranked from 1 to 10 (weakest to strongest). In Experiment 1, we found a strong positive correlation between intensity and amplitude (r = 0.2, p=$2.1 \times 10^{-5}$, Pearson linear correlation coefficient), with an intensity of 2.4 ± 1.9 a.u. (mean ±s.d.) for 20 µA and 4.0 ± 2.1 a.u. for 100 µA, with a slope of 0.02 (1st-order polynomial, least squares fitting). As subjective measures of intensity are most likely sensitive to day-to-day variability, in post-hoc analysis we also normalized intensity values within each session (see Materials and methods). We measured a negative correlation between the current amplitude and the standard deviation of the intensity (r = −0.6, p=0.12). Duration of the percept was recorded for each response as either *short* (sensation lasts only briefly at the onset of stimulation), *medium* (sensation persists throughout the stimulation but not for the full length of the stimulation) or *long* (sensation lasts the full duration of the stimulation). The majority of responses were short (N = 225), followed by medium (N = 122) with very few long responses (N = 12). Stimulus duration was not recorded for 22 responses of the 381 responses. For Experiment 2 this trend was replicated (N = 268, 55 and 1. Short, medium and long, respectively). There was no relationship between duration of the sensation and either amplitude of stimulation (p=0.1, $\chi^2$(*1323*)=*4.53*) or frequency of stimulation (p=0.2, $\chi^2$(*1323*)=*2.83*).

To our knowledge, this is the first report in human of replicable, purely naturalistic proprioceptive and cutaneous sensations induced through ICMS. Stimulation over a wide range of amplitudes and frequencies generated qualitatively diverse sensations, although percept modality was strongly linked to variations in amplitude. Pairing these natural sensations with BMIs create a unique opportunity to explore how effectively they can be incorporated in a closed-loop BMI system. For example, the ability to evoke proprioceptive sensations could allow the subject to interpret position- or movement-related information, as previously reported in primate studies (*Tomlinson and Miller, 2016*; *Dadarlat et al., 2015*), while eliciting cutaneous sensations could improve our ability to deliver richer somatosensory feedback for object manipulation. Together these somatosensory signals have the potential to improve performance and embodiment when using a BMI-controlled external device.

## Materials and methods

### Subject

We recruited and consented a 32-year-old male participant (FG) with C5-level complete spinal cord injury, 1.5 years post-injury, to participate in a clinical trial of a BMI system with intracortical recording and stimulation. The subject has residual sensation in the anterior-radial section of his upper arm, and some residual sensation in the posterior-radial section of his upper arm and forearm, which present as paresthesias. All procedures were approved by the Institutional Review Boards (IRB) of the University of Southern California (USC) and Rancho Los Amigos National Rehabilitation Hospital (RLA). The implant procedure occurred at Keck Hospital of USC, and study sessions took place at RLA.

### Surgical planning and implantation

Surgical planning followed the protocols described in (*Aflalo et al., 2015*), with an additional task for identifying an implant location within somatosensory cortex. In this task, a visual cue prompted the experimenter, who was standing next to the MRI, to reach into the MRI machine with a wooden pole and repeatedly press at one of three points on the subjects right upper limb where he previously reported residual paresthetic sensation; biceps, forearm and thenar eminence. The subject was instructed to attend to any residual sensation he felt at each location and report the number of times the experimenter touched him on the cued location (*Kastner et al., 1998*; *Staines et al., 2002*). After functional imaging, three target locations for electrode placement were identified; supramarginal gyrus (SMG), ventral premotor cortex (PMv) and primary somatosensory cortex (S1). One 96-channel, platinum-tipped Neuroport microelectrode recording array (Blackrock Microsystems, Salt Lake City, UT) was implanted in each of SMG and PMv. Two 7 × 7 SIROF (sputtered iridium oxide

film)-tipped microelectrode arrays (with 48 physically-connected channels each) were implanted in S1. The SIROF-tipped electrodes have lower impedance than the platinum-tipped electrodes, and thus are better suited to stimulation.

## Stimulation and recording parameters

All stimuli consisted of biphasic, charge-balanced, cathodic-leading pulses, with 200 μs width per phase, 53 μs interphase interval, and one-second stimulus duration delivered to a single electrode on the S1 array only. The maximum charge delivered per phase was 20 nC. We selected these parameters, and set electric charge limits according to safe ranges shown in ICMS studies with NHPs (*Kim et al., 2015a*). Stimulation was delivered with a Blackrock CereStim device, and stimulation parameters were set and delivered using the CereStim API through MATLAB (The Mathworks Inc, Natick, MA) software (MATLAB code in Source code file 1).

## Task

Experiment 1: After initial assessment of implant viability, we evaluated the effects of stimulation parameters through a percept-detection task. For this primary mapping task, each of the 96 stimulation electrodes were evaluated at eight amplitudes: 20, 30, 40, 60, 70, 80, 90, and 100 μA, at 150 Hz. The subject was seated in a wheelchair approximately 1.5 meters from a TV screen. The subject was instructed to look at a fixation point in the middle of the screen throughout the experiment. In each trial, after a three-second inter-trial interval, the subject was presented with a large purple circle on the screen indicating that an electric stimulus was being delivered. Then, after a one-second delay, an auditory cue signaled the subject to report whether he felt any sensation. When a sensation was perceived, the subject reported its location on a body and hand map, with anterior and posterior views, by referencing a fine overlaying grid (*Figure 1*). The subject also reported qualitative characteristics including the perceived stimulus intensity, the perceived duration of stimulation, and a description of the sensation (*Table 1*). Sensations closer in nature to tactile stimuli were classified as *cutaneous*, and those triggering a feeling of movement or change in position were classified as *proprioceptive*. To complete the mapping of amplitude, we ran trial blocks where we randomly selected a subset of electrodes. Each block contained three replicates of stimulation per parameter, per electrode. An additional set of trials, numbering 10% of the total trials in a block, were added as 'catch' trials, where the visual stimuli on the screen and auditory response cue remained identical but the stimulation did not occur. Catch trials were randomly interleaved among the normal trials. In each block, trials were ordered such that stimulation did not occur to the same or adjacent electrodes concurrently.

Experiment 2: For the second mapping task, five electrodes were selected for further evaluation at different amplitudes and frequencies. All the phases of the task and other stimulation parameters were the same as in the previous mapping task. The subset of electrodes selected for this task were those that exhibited the most reliable responses in the first mapping task. We varied the current amplitude (20, 40, 60, 80, 100 μA) and pulse frequency (50, 100, 150, 200, 250, 300 Hz), and tested each amplitude-frequency combination six times per electrode. The full dataset was obtained over three consecutive days. In each day, each of the five electrodes received two replicates of all possible amplitude and frequency combinations. The order of electrode stimulation was determined pseudorandomly.

## Statistics and analysis methods

Throughout the analysis we used the Kruskal-Wallis nonparametric ANOVA statistical test. We calculated correlations between responses using the Pearson linear correlation coefficient.

To examine whether response history had a significant effect on the proportion of reported sensations (*de Lafuente and Romo, 2005*), we looked at differences between the distribution of reported sensations during stimulation for three conditions: all trials, trials after a reported sensation (*hit*) and trials after no reported sensation (*miss*). We estimated these distributions for each amplitude in a given experimental session across all tested electrodes, and used Kruskal-Wallis nonparametric ANOVA with Dunn-Sidak multiple comparisons correction to test for significance at each amplitude. Furthermore, we generated a shuffle distribution of probabilities with N = 10,000 permutations for hits following a hit or a miss for each amplitude. We found no significant difference

between the shuffle distributions and the empirical data, with the actual proportion being within the 5th-95th percentile range of the shuffle distribution. For the bootstrapped resampling of proprioceptive and cutaneous responses in Experiment 1, we drew 15 samples at each iteration from the total responses at each amplitude (range 21–93 responses across all amplitudes). Where normalized intensity data are reported, we rescaled the raw intensity (range 1–10) to a normalized scale (range 0–1) for each day by subtracting the minimum and then dividing by the maximum.

Raw data for all analysis presented in this manuscript can be found as downloadable source data 'Responses to single-electrode stimulation'. Specific details can also be found in the first sheet of the raw data file.

## Acknowledgements

We would like to thank FG for his efforts and engagement in the clinical study, and the clinical staff at Rancho Los Amigos for their work and dedication during the experimental sessions.

## Additional information

### Funding

| Funder | Grant reference number | Author |
|---|---|---|
| National Institute of Neurological Disorders and Stroke | 5U01NS098975-02 | Michelle Armenta Salas<br>Luke Bashford<br>Spencer Kellis<br>Kelsie Pejsa<br>Brian Lee<br>Charles Y Liu<br>Richard A Andersen |
| Della Martin Foundation | | Michelle Armenta Salas |
| David Geffen School of Medicine, University of California, Los Angeles | David Geffen Medical Scholarship | Matiar Jafari |
| James G. Boswell Foundation | | HyeongChan Jo<br>Richard A Andersen |
| National Science Foundation | 1028725 | HyeongChan Jo |
| National Institute of Neurological Disorders and Stroke | NS099008-01 | Daniel Kramer |
| T & C Chen Brain-machine Interface Center | | Richard A Andersen |

The funders had no role in study design, data collection and interpretation, or the decision to submit the work for publication.

### Author contributions

Michelle Armenta Salas, Luke Bashford, Resources, Data curation, Software, Formal analysis, Validation, Investigation, Visualization, Methodology, Writing—original draft, Writing—review and editing; Spencer Kellis, Conceptualization, Resources, Data curation, Software, Formal analysis, Supervision, Funding acquisition, Validation, Investigation, Visualization, Methodology, Writing—original draft, Writing—review and editing; Matiar Jafari, Software, Investigation, Methodology; HyeongChan Jo, Methodology; Daniel Kramer, Resources, Investigation; Kathleen Shanfield, Resources; Kelsie Pejsa, Resources, Project administration; Brian Lee, Resources, Methodology; Charles Y Liu, Conceptualization, Resources, Funding acquisition, Methodology; Richard A Andersen, Conceptualization, Resources, Supervision, Funding acquisition, Validation, Methodology, Project administration, Writing—review and editing

## Author ORCIDs

Michelle Armenta Salas  http://orcid.org/0000-0002-0634-2891
Luke Bashford  http://orcid.org/0000-0003-4391-2491
Spencer Kellis  http://orcid.org/0000-0002-5158-1058
Richard A Andersen  http://orcid.org/0000-0002-7947-0472

## Ethics

Clinical trial registration: NCT01964261
Human subjects: This study was conducted in accordance with a protocol reviewed and approved by the FDA as well as Institutional Review Boards at Rancho Los Amigos National Rehabilitation Center and the University of Southern California (associated protocol numbers: Caltech IRB #15-0501, USC IRB #HS-13-00492 and RLA IRB #154). The subject provided informed consent to participate in the study, and also gave informed consent to publish.

## Decision letter and Author response

Decision letter https://doi.org/10.7554/eLife.32904.010
Author response https://doi.org/10.7554/eLife.32904.011

## Additional files

### Supplementary files

• Source code 1. Stimulation commands.
DOI: https://doi.org/10.7554/eLife.32904.007

• Transparent reporting form
DOI: https://doi.org/10.7554/eLife.32904.008

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
