## [Decision Letter]

Thank you for submitting your article "Proprioceptive and Cutaneous Sensations in Humans Elicited by Intracortical Microstimulation" for consideration by *eLife*. Your article has been reviewed by three peer reviewers, one of whom, Ranulfo Romo is a member of our Board of Reviewing Editors and the evaluation has been overseen by Richard Ivry as Senior Editor. The following individuals involved in review of your submission have agreed to reveal their identity: Victor de Lafuente (Reviewer #2); Apostolos P Georgopoulos (Reviewer #3).

The reviewers have discussed the reviews with one another and we have worked together to draft this decision letter to help you prepare a revised submission.

Summary:

The reviewers found your paper contains important results for understanding the role of human somatosensory cortex and brain-machine interface. The reviewers have a number of recommendations for revision, summarized below in terms of major and minor issues.

1) It would be helpful for the authors to explore the effects of trial history on the subject's reports. Similar to de Lafuente and Romo, 2005 (Figure 3; although we recognize that are no trials equivalent to "false alarms"), certain trial outcomes might skew the outcome probabilities in the following trial. For example, it could be the case that successful stimulation (which leads to percept formation) increases the chances of future positive reports. An analysis of this kind will help make an even more complete presentation of the results, and its absence would be noted by future readers.

2) You do not explain the results obtained by microstimulating the PMv and the SMG arrays. Did stimulation with these arrays fail to elicit any sensation? Why would that be case? Please provide a discussion on this issue.

3) Were recordings made from the implanted electrodes (we assume so)? Assuming so, we expect there would be useful information in these data, such as the number of recorded neurons, their basal activity, and how activity changed across the four weeks of the stimulation protocol. This information would be useful in future studies that extend your work.

4) The authors mention that previous ICMS protocols have elicited tingling and buzzing sensations in the patients (Introduction). What was different about the current protocol that led to the reports of more natural sensations?

---

## [Author Response]

Summary:The reviewers found your paper contains important results for understanding the role of human somatosensory cortex and brain-machine interface. The reviewers have a number of recommendations for revision, summarized below in terms of major and minor issues.1) It would be helpful for the authors to explore the effects of trial history on the subject's reports. Similar to de Lafuente and Romo, 2005 (Figure 3; although we recognize that are no trials equivalent to "false alarms"), certain trial outcomes might skew the outcome probabilities in the following trial. For example, it could be the case that successful stimulation (which leads to percept formation) increases the chances of future positive reports. An analysis of this kind will help make an even more complete presentation of the results, and its absence would be noted by future readers.

We examined whether a hit (response to stimulation) or a miss (no response to stimulation) altered the probability of a response in the following trial. First, we followed a method similar to de Lafuente and Romo (2005) (Figure R1A). At each amplitude, we tested for significant differences between these populations, and found a significant effect only for 30 μA. However, methodological differences could affect the above analysis. For example, in our experiments electrodes were pseudorandomly interleaved in blocks of trials, so that subsequent trials stimulated different electrodes and the number of trials between stimulation at the same electrode varied over time. In the case of the 30 μA noted above, there were fewer electrodes for which stimulation generated a response, meaning that the proportion of “hits following hits” was much lower.

To address these differences, we compared the empirical probabilities of hits following a hit and hits following a miss to the same probabilities calculated from shuffling the data at each amplitude with ten thousand repetitions. We tested the difference between the shuffle distribution of response proportions and the actual proportion at each amplitude. We found the actual value to be within the 5th-95th percentile of the shuffle distribution for each amplitude (Figure R1B), and when pooling across amplitudes. These results indicate that any effect of trial history is not significantly different to chance.

We have written additional text in the manuscript to present this result and updated the “Statistics and Analysis Methods” section.

Results and Discussion sections: “No false positives were reported in any catch trials, and we found no effect of trial history in the proportion of reported sensations during stimulation (see Materials and methods section).”

Subsection “Statistics and Analysis Methods”: “To verify whether response history had a significant effect on the proportion of reported sensations (de Lafuente and Romo, 2005), we looked at differences between the distribution of reported sensations during stimulation for three conditions: all trials, trials after a reported sensation (hit) and trials after no reported sensation (miss). We estimated these distributions for each amplitude in a given experimental session across all tested electrodes and used Kruskal-Wallis nonparametric ANOVA with Dunn-Sidak multiple comparisons correction to test for significance at each amplitude. Furthermore, we generated a shuffle distribution of probabilities with N = 10,000 permutations for hits following a hit or a miss for each amplitude. We found no significant difference between the shuffle distributions and the empirical data, with the actual proportion being within the 5th-95th percentile range of the shuffle distribution.”

2) You do not explain the results obtained by microstimulating the PMv and the SMG arrays. Did stimulation with these arrays fail to elicit any sensation? Why would that be case? Please provide a discussion on this issue.

For these data, only the electrodes in somatosensory cortex were stimulated. Our experimental protocol allows for recording, but not stimulation, through the electrodes in PMV and SMG. To clarify this in the text we have added additional phrasing to several sentences:

Introduction: “obtained from stimulation in S1 of a single human”

Subsection “Stimulation and Recording Parameters”: “delivered to a single electrode on the S1 array only.”

3) Were recordings made from the implanted electrodes (we assume so)? Assuming so, we expect there would be useful information in these data, such as the number of recorded neurons, their basal activity, and how activity changed across the four weeks of the stimulation protocol. This information would be useful in future studies that extend your work.

Under our experimental protocol, we did not record from the S1 electrodes, but we did take impedance measurements at the beginning of each session. Recording to address the effects of stimulation have been made for subsequent studies, but we consider these data and experiments outside the scope of the present manuscript.

4) The authors mention that previous ICMS protocols have elicited tingling and buzzing sensations in the patients (Introduction). What was different about the current protocol that led to the reports of more natural sensations?

With regards to the only other human ICMS study (Flesher et al., 2016), our protocol was similar in stimulus duration, charge balancing, leading polarity, current amplitude and pulse frequency ranges. Our protocol diverged from that study in pulse symmetry, phase duration and interphase interval, and we used a different behavioral protocol. Other factors such as array implant location, time after injury, and level of spinal cord injury vary across subjects and are likely to contribute to reported differences. Because these factors are difficult to control experimentally, our protocol, like the above-referenced study, was designed so that the subject would serve as his own control. Thus, we consider results from both of these studies valid in context and look forward to future studies investigating the relationship between these many factors and the qualitative nature of the elicited percepts.